# Comparison of 12 Different Animal Welfare Labeling Schemes in the Pig Sector

**DOI:** 10.3390/ani11082430

**Published:** 2021-08-18

**Authors:** Katriina Heinola, Tiina Kauppinen, Jarkko K. Niemi, Essi Wallenius, Satu Raussi

**Affiliations:** 1Bioeconomy and Environment, Natural Resources Institute Finland (Luke), Latokartanonkaari 9, 00790 Helsinki, Finland; katriina.heinola@luke.fi; 2The Finnish Centre for Animal Welfare, Natural Resources Institute Finland (Luke), 00790 Helsinki, Finland; tiina.kauppinen@luke.fi; 3Bioeconomy and Environment, Natural Resources Institute Finland (Luke), 60320 Seinäjoki, Finland; jarkko.niemi@luke.fi; 4Department of Production Animal Medicine, Faculty of Veterinary Medicine, University of Helsinki, 00014 Helsinki, Finland; essi.wallenius@armentabenessi.com

**Keywords:** animal welfare, label, pigs, animal welfare scheme

## Abstract

**Simple Summary:**

Welfare requirements from an animal point of view are the same regardless of the country. However, differing requirements of animal welfare schemes make it hard for consumers to make informed choices. Therefore, an open and coherent labeling scheme that provides information on farm animal welfare will be beneficial from the consumer perspective. We reviewed 12 pig welfare schemes. We aimed to identify consistencies and differences in welfare requirements between these schemes. The studied welfare requirements were heterogeneous in the potential each scheme had to advance pig welfare. Certain requirements barely exceeded the minimum standards for the protection of pigs in European Union (EU) legislation, but the more demanding tiers of multitier schemes had the potential to enhance animal welfare. The most ambitious tiers could improve animal welfare substantially and, in terms of resources available to the animal, they often were convergent with organic animal farming standards. Because of variation of welfare requirements between the labels, it was challenging to compare existing labeling schemes. Adopting a harmonized labeling terminology and standard, increased use of animal-based measures, and open communication will make labeling more reliable and transparent, which will contribute to the availability of standardized animal-friendly products and will be equitable from an animal welfare perspective.

**Abstract:**

Animal welfare labeling schemes have been developed to respond to consumers’ expectations regarding farm animal welfare. They are designed to certify that labeled products comply with certain animal welfare standards. In this study, 12 pig welfare labeling schemes were reviewed, and their criteria related to pig welfare were compared. Information regarding farrowing criteria, space allowance, outdoor access, mutilations, and provision of enrichments and bedding material were gathered from the labels’ internet pages and documentation. The results indicated a substantial variation between the labels in terms of the level of animal welfare they ensure. While certain schemes barely exceeded the minimum standards for the protection of pigs in the European Union, more demanding tiers of the multitier schemes had the potential to improve animal welfare substantially. The most ambitious tiers of multistage schemes were often comparable to organic standards providing outdoor facilities and additional space. The heterogeneity of the labels’ standards complicates the comparison of labels.

## 1. Introduction

Animal welfare can be defined as an animal’s experience of its own mental and physical state [1]. The welfare of pigs in the European Union (EU) is regulated by the Council Directive 2008/120/EC on the minimum standards for the protection of pigs [2]. However, European consumers are increasingly focused on farm animal welfare (FAW) and expect improvements in FAW [3]. Many consumers are also uncertain about whether animal welfare standards are met in livestock production [4]. Hence, new FAW labeling schemes have been introduced into the markets in recent years. For instance, two different labels were launched in Denmark within a short time period, “Animal welfare hearts” by Coop retailer in 2016, and Bedre Dyrevelfærd by the Danish Government in 2017 [5]. Individual member states in the European Union and in other European countries have also been active in creating national labeling initiatives. An EU-wide FAW labeling has been proposed, which will harmonize the criteria of different welfare labeling schemes [6,7]. Until now, there is no EU-wide welfare label, and the level of welfare ensured by different labeling schemes may thus vary substantially from case to case.

The EU requirements for organic animal production [8] aim to provide an elevated level of animal welfare by ensuring that there are more resources at the disposal of the animals than in conventional production, e.g., reduced stocking density and outdoor access. However, because organic production addresses sustainability in a broad sense versus specifically addressing animal welfare, organic production leaves room for more dedicated FAW labeling schemes. Organic legislation in the European Union is subject to change and new organic legislation will enter into force in the EU on 01 January 2022.

A FAW labeling scheme is designed to certify that labeled products comply with certain FAW standards. FAW labeling may facilitate the transition of the pig industry from volume-based to concept-based production, where additional value is sought by supplying quality attributes of production [9]. Van Loo et al. [10] studied different sustainability labels of meat and found that free-range claims were the most frequently preferred claims among consumers. They also noted that an EU-level FAW label would be warranted. The consumer demand for premium products is an essential driver for labeled products. However, because there is limited awareness of labels among consumers, the wide range of different sustainability labels offering improvements in FAW and in environmental characteristics of production may be confusing [11]. Furthermore, Heerwagen et al. [12] concluded that to improve FAW, it is important to distinguish those products with strengthened FAW from products with only a medium level of welfare enhancements.

The meta-analysis by Janssen et al. [13] suggested that for consumers, the most important FAW-related factors were outdoor access, stocking density, floor type, and naturalness in general, which refers to the freedom to allow an animal to behave according to its’ natural instincts. Basic needs, such as access to food and water, were considered important by citizens, but also issues in transport and slaughter process are essential [14]. The features mentioned above represent resources given to animals to allow for a state of good welfare. Resource-based measures can be considered as prerequisites for good welfare, but they do not necessarily guarantee a good welfare state per se. Animal-based measures, such as the absence of shoulder damages of sows or aggressive behavior observed between group-housed pigs, are more challenging to communicate to the markets and more challenging to monitor and ensure than resource-based measures. However, animal-based measures are often related to the experience of an individual animal, and thus animal welfare more directly than resource-based measures [15].

While consumer demand for and the concerns of FAW offer opportunities for developing labels, these do not always translate into actual purchasing behavior. For instance, price premiums asking for high-end products and concerns regarding the reliability of labeling schemes may reduce public interest toward labeled products [14]. The public has concerns regarding intensive production systems in relation to FAW, naturalness, and the use of antibiotics, and they prefer proactive herd management approaches. It is imperative to address the concerns that the public may hold, in both communication and policy strategies. This will require transparent, coordinated, and trustful communication between producers and the public [4].

A European consumer can purchase pork produced in or outside of Europe. The minimum animal welfare requirements are set at both the national and EU-level, which challenges the knowledge of the consumer and the transparency of production. A FAW label can be an important vehicle for a trusted communication. A FAW label can both inform consumers about the country’s level of FAW and provide the basis for common FAW standards covering both domestically produced goods and goods imported from outside the EU [16]. Moreover, an EU-wide harmonized FAW label can contribute to the development of an EU single market for animal-friendly products.

While there are several FAW labeling schemes on the market, there is more to investigate on how well they comply with each other’s standards. Hence, the objective of this study was to analyze the characteristics of FAW labeling schemes for pigs (scheme intercomparison). The pig production criteria and compliance of 12 selected FAW labeling schemes was compared with the minimum standards of the EU requirements for the protection of pigs [2] and of organic production [8]. While this study did not conduct consumer research, it examined the accessibility of information about labeling systems and how understandable the added value of labeling for the welfare of pigs is.

## 2. Materials and Methods

### 2.1. Pig Welfare Labels Studied

Based on an internet and document search, 12 generic animal welfare labeling schemes were selected for the review. Possible animal labels were first identified by searching for information on different animal welfare labeling schemes that were on the market in the EU or North America or were in preparation in 2017. An identified generic labeling scheme was included in further analysis when sufficient transparent information on the indicators of pig welfare that were studied was available. Nine schemes were operating in Europe and three in North America (Table 1). As pork is exported worldwide, the labels were selected from both European countries and from North America.

The EU has common minimum standards for the welfare of pigs [2] as well as for organic production [8], and existing labels were compared to these standards. The following labels were considered in this study (for clarity of presentation, references to specific labels are provided only once, upon the introduction of the scheme): Für Mehr Tierschutz (Bonn, Germany, for finishing pigs only), Staatliches Tierwohlkennzeichen (a FAW label currently in preparation by the Federal Ministry of Food and Agriculture in German Das Bundesministerium für Ernährung und Landwirtschaft BMEL in Bonn Germany), Beter Leven, Bedre Dyrevelfærd, Coop (Denmark and Switzerland), Friland free range (Denmark), RPSCA assured, Animal Welfare Approved (AWA, Terrebonne, OR, USA), Humane Farm Animal Care (Certified Humane, HFAC, Middleburg, VA, USA), and Global Animal Partnership (GAP, Austin, TX, USA).

In Europe, the Dutch label Beter Leven was launched in 2007 by the Dutch Society for the Protection of Animals and it considers multiple farm animal species in addition to pigs [17]. The criteria are set for three tiers, representing three levels of ambition regarding FAW. The Danish label Friland was established in 1992 as an “organic first mover” and Friland’s free range line was reviewed here (separate from the Friland Organic) [18]. Coop Denmark’s “Animal welfare heart” label was introduced in 2016 with four welfare tiers [19]; here, the focus is on tiers 1–3 only (on the 4th tier, animal welfare must exceed the requirements of the 3^rd^ tier substantially). Bedre Dyrevelfærd is a three-tiers label by the Danish government [20]. The German Animal welfare label in preparation by BMEL has three tiers and is also a governmental labeling scheme [21]. The Für Mehr Tierschutz label by the German animal welfare association, Deutscher Tierschutzbund, for finishing pigs (to date) was also included in the review [22]. The RSPCA Assured (previously Freedom Food) label was established in Great Britain in 2015 [23]. However, the charity (the Royal Society for the Prevention of Cruelty to Animals) has a history that predates the establishment of the label. Coop Switzerland promotes the Naturafarm [24] and organic Naturaplan [25] labels. The FAW-focused Naturafarm label was established to distinguish from the purely organic label Naturaplan.

Animal Welfare Approved (AWA) [26], Certified Humane (HFAC) [27], and Global Animal Partnership (GAP) [28] were established in the United States, but they are present around the world. GAP by Whole Foods Market was established in 2008 and is operated by an independent non-profit organization. It has five tiers and an additional plus tier for the fifth tier. The second tier is (enriched) with enrichments, the third requires outdoor access, the fourth requires a pasture for at least the majority of an animal’s life, and the fifth is completely animal-centered, which means, for example, that animals are always on the pasture, on the same farm, and physical alterations are not allowed. AWA, managed by Greener World, requires pasture-based animal husbandry. Humane Farm Animal Care (HFAC) is an international non-profit certification organization established in 1998, covering multiple farm animal productions. Their label is Certified Humane, which focuses on kind and more responsible animal husbandry practices.

Information about the characteristics of the labels was retrieved from their websites and published documents. The welfare requirements and information were gathered in 2020. Missing criteria was enquired by contacting the responsible organization. The data included the organization responsible for the label, the year the scheme was established, the inspection method, inspection frequency, and body inspecting the farms or the scheme, the number of tiers, and the characterization of requirements that the label had on animal welfare. The information was compared to the EU standards for the protection of pigs [2] and for organic production in the EU [8]. In addition, the criteria were compared between the schemes.

### 2.2. Animal Welfare Implications of Measures Covered by the Labels

Based on the selected labels, the following FAW-related production attributes were considered in the analysis: space allowance of fattening pigs, the use of bedding and enrichments, outdoor access, the prohibition of farrowing crates, the duration of the suckling period, the type of farrowing space (including the space allowance for minimum total space and lying area), the prohibition of mutilations such as tail docking, teeth clipping, or castration, and the maximum duration on animal transportation. This section summarizes welfare considerations related to these attributes.

Typical requirements for pig welfare labeling schemes include free-farrowing of sows, longer-than-usual minimum suckling time, more space, bedding, and enrichments than usual, and the avoidance of painful procedures such as tail docking and surgical castration. Limiting slaughter transport time to a certain maximum number of hours is also included in many welfare labeling schemes. In the Council Regulation (EC) No 1/2005 [29] on the protection of animals during transport, more than eight hours is classified as long transport of animals.

#### 2.2.1. Free Farrowing

Confining a farrowing sow into a cage impairs her welfare compared to free far-rowing by preventing the sow from fulfilling its behavioral needs, especially nest building behavior prior to farrowing [30,31].

#### 2.2.2. Space Allowance

Skin score and growth rate of medium-sized pigs are improved when the space is increased by 0.8 m^2^ per pig to 1 m^2^ per pig [32,33]. Munsterhjelm et al. [34] found a de-crease of tail lesions (among pigs raised with intact tails) in up to at least 1.5 m^2^ per fattening pig.

Sows benefit from greater space allowance. Weng et al. [35] found that aggressive behavior and skin lesions decreased and rooting behavior increased when pregnant sows were provided with a pen area of 4.8 m^2^ in comparison to 2, 2.4 or 3.6 m^2^ per sow in a group pen. In addition to pen area, sow aggression is affected by feeding routines and feeding system [36].

#### 2.2.3. Outdoor Access

Outdoor and pasture access enables pigs to exhibit natural behaviors such as rooting [37]. However, outdoor access may also increase the prevalence of leg disorders [38,39].

#### 2.2.4. Enrichments

Exploring the environment by chewing is inherent behavior of pigs [40]. Therefore, it is important for the pigs’ welfare to provide enrichments to explore [33,41,42].

#### 2.2.5. Bedding

Bedding is an important factor for both fattening pigs’ and sows’ welfare [34]. In fattening pigs, bedding has decreased fights and tail biting, and in sows it has reduced frustration and bursitis [34].

#### 2.2.6. Painful Procedures

During rearing, pigs can undergo several painful procedures such as tail docking, teeth clipping, or surgical castration. These treatments place pigs at risk for pain, tissue damage, and infections [43]. For example, EU legislation on the protection of pigs [2] allows for the surgical castration of male piglets without analgesia, provided they are under seven days old.

## 3. Results

### 3.1. Inspection of the Schemes

The control of the schemes is essential from the perspective of the schemes’ credibility. The control of farmers’ husbandry practices and outcomes and the scheme is typically conducted by an independent and accredited body (Table 1). The inspection is performed annually for the schemes examined, except for GAP, which maintains the certificate validity for 15 months. Information available for Naturafarm defined regular inspections and Für Mehr Tierschutz mentioned risk-based control scheme. Beter Leven had their own separate certifying foundation. Friland, Naturafarm, and the RSPCA defined designated auditors.

### 3.2. Measures to Promote the Welfare of Pigs in the Schemes

When comparing labeling schemes, this study focused on the space requirement of fattening pigs, the use of bedding and enrichments, and outdoor access. Regarding farrowing, the use of farrowing crates, length of the suckling period and type of far-rowing space, including the space allowances for minimum total space and lying area, was examined. In addition, allowing mutilations such as tail docking, teeth clipping, and castration were included, as well as the transport time requirement.

#### 3.2.1. The Space Requirements and Outdoor Access for Fattening Pigs

The Council Directive for the protection of pigs requires indoor space of 0.65 m^2^ and 1 m^2^ for pigs weighing 85–110 kg and over 110 kg, respectively [2]. The EU organic legislation requires 1.3 and 1.5 m^2^ space for pigs of 86–110 kg and over 110 kg, respectively, and outdoor access from May to October with a minimum of 1 m^2^ outdoor area per pig [44].

Considering the minimum space allowance for growing pigs weighing no more than 110 kg, the minimum total space allowances of the FAW schemes were the lowest in Bedre Dyrevelfærd tier 1 (Bedre Dyrevelfærd tier 1 did not define the space requirement but mentioned that the space must be more than required in the order on the protection of pigs “depending on the individual herd’s specific production organization, including the requirement that does not permit tail docking”) and the RSPCA (0.75 m^2^). However, the weight classification varied between the labels as shown in Figure 1. The space requirement in AWA and tier 3 of Beter Leven and Coop DK reached the space allowance of the EU organic regulation (1.3 m^2^). Naturaplan’s requirement of indoor area was 1 m^2^ per pig (Naturaplan required minimum total stable area 1.65 m^2^, including 0.6 m^2^ lying area and 0.65 m^2^ outdoor area (0.33 m^2^ unroofed open area)).

There were no requirements for bedded lying areas in the Beter Leven, Coop DK, Für Mehr Tierschutz, and BMEL label for tiers 1 and in Bedre Dyrevelfærd for tiers 1 and 2. When required, the area varied from 0.5 to 1.3 m^2^ per pig (Figure 1). Beter Leven tiers 2 and 3 required floors to be bedded with straw or similar material and 50% floors to be solid. The minimum lying area requirements were larger in the North American schemes than in the European, but the total space allowance was, in general, smaller. The lying area in GAP tiers 3, 4, and 5 was 0.84 m^2^ indoors, but this included continuous outdoor access. Hence, the space allowances were not entirely comparable.

Outdoor access was required by Friland free range, AWA, Naturaplan, and Naturafarm, tiers 2 and 3 of Beter Leven and Coop DK, tier 3 of Bedre Dyrevelfærd and BMEL’s label, and by GAP tiers 3, 4, and 5. The schemes requiring bedded indoor area also required outdoor access, with the exceptions of BMEL tier 2, the RSPCA, and HFAC, and GAP tiers 1 and 2, which did not require outdoor access but required a lying area. GAP tier 4 and 5 and AWA required pasture access.

The first tiers of the multitier schemes did not require outdoor access for fattening pigs. AWA’s outdoor area must be at least 5.2 m^2^. GAP tier 4 (additional area of 0.84 m^2^, of which 0.56 m^2^ must be outdoors and the remaining space can be indoors or outdoors) required an area of 0.84 m^2^ in addition to the lying area for the period if the pig needed to be removed from the pasture. There were no requirements for the pasture area at GAP tier 5. However, earlier GAP required the corresponding outdoor area versus bedded lying area indoors. There were also specific outdoor requirements for certain animal groups, for example Beter Leven required pasture access for gestating sows and Naturaplan for dry sows.

#### 3.2.2. Floor Type, Bedding, and Enrichments

The requirement for solid floor along the slatted floor was defined by all the schemes (Table 2) that required lying space; however, this information was not available in Bedre Dyrevelfærd tier 3. In general, the required lying area must minimally be on solid floor. Beter Leven tier 1, which did not define the minimum lying area, required a solid floor with a minimum of 40% of the total area [17], while Für Mehr Tierschutz tier 1 allowed fully slatted floors. Bedding was required by all the schemes (Table 2) except for Für Mehr Tierschutz tier 1. Despite the requirements, there were variations in the quality of the bedding. The RSPCA and HFAC required bedding to a “sufficient extent to avoid discomfort”. Straw or equivalent materials were required in Beter Leven tiers 2 and 3, in Bedre Dyrevelfærd for tier 3, Coop DK’s, Friland, BMEL’s label tier 2, Naturaplan and Naturafarm, and AWA and GAP.

All the schemes except GAP in tier 1 required enrichment materials (Table 2). GAP tier 5 did not mention enrichments, but the pigs were kept outside in the pasture at all times. Again, the quality and the amount of enrichment varied. Straw was recommended by most schemes, but also other materials and manipulable objects, such as hay, silage, wood shavings branches, whole crop peas or barley, compost, peat, sisal ropes, or other natural materials were mentioned.

#### 3.2.3. Farrowing-Related Criteria

The use of farrowing crates is prohibited in organic production [8] but allowed in conventional production in the EU [2]. Farrowing-related criteria are summarized in Figure 2. Farrowing crates were allowed by few welfare schemes. Neither of the German labels considered farrowing issues as Germany is to update the legal requirements related to farrowing. Beter Leven allowed farrowing crates at tier 1, and with “openable pen walls” (sides) for five days and three days at tiers 2 and 3, respectively. Bedre Dyrevelfærd also allowed the use of sides for four and two days at tiers 1 and 2, respectively. The use of crates was prohibited in Friland, Bedre Dyrevelfærd tier 3, Co-op’s “Welfare Heart” tiers 2 and 3, GAP, AWA, HFAC, and in Für Mehr Tierschutz tier 2. HFAC prohibited the use of “traditional straight, narrow farrowing crates”.

There were requirements for the farrowing place in terms of the minimum total area and the minimum (bedded) lying area. Both definitions were available for GAP, Naturaplan, Naturafarm, and the RSPCA. Sows must be mainly able to farrow in the outdoor huts in Coop DK’s tiers 2 and 3, in Friland, AWA, and in GAP tier 5.

The length of suckling period, when defined, ranged from 23 days up to 56 days, being highest in the GAP for tier 5. The comparable period to organic production, the minimum of 40 days, was required in GAP for tiers 3 and 4 (42 days) and 5, AWA (42 days), Naturaplan (42 days), and in Coop DK’s for tier 3 (40 days). Although in the EU, piglets must not be weaned before 28 days of age; there are exceptions for the shorter period, only 21 days of age, when piglets are moved to specialized housing [2].

In addition to farrowing place and minimum weaning age, the provision of nest building material before farrowing and confining sows into farrowing pen were also defined by certain schemes. Bedre Dyrevelfærd tier 1 allowed the movement of sows from groups to the farrowing pen no more than seven days before expected farrowing. In tier 3, sows were allowed to be placed in huts no more than five days before expected farrowing. The same five days applied for Friland. Coop tier 1 defined the time at 3–5 days. AWA defined that “sows must not be placed into individual pens for more than two weeks prior to the expected farrowing date” and they cannot be confined within huts for more than 24 h before expected farrowing.

The provision of nest building material was mentioned by Beter Leven, Bedre Dyrevelfærd, Naturafarm, the RSPCA, HFAC, and GAP. Beter Leven and HFAC required material, at the latest, 48 h before farrowing, as well as the RSPCA (with the definition of at least 2 kg straw). GAP, Bedre Dyrevelfærd, and Naturafarm required providing nest building material 3, 5, and 2–4 days before farrowing, respectively.

#### 3.2.4. Transport Time

The Council regulation on the protection of animals during transportation [29] specifies that pigs can be transported for 24 h, after which a rest period of 12–24 h must be taken. After this, the corresponding transport and rest periods can continue again. The transport guidelines for organic animal production in the EU follow the provisions of the Council regulation on the protection of animals during transportation [29]. The maximum transport time, from farm to slaughterhouse, was eight hours by all the schemes, except for Für Mehr Tierschutz, which allowed only four hours transportation and GAP which allowed 16 h transportation at tiers 1–5. The additional 5+ tier prohibited the transportation of pigs, instead, pigs are slaughtered on the farm.

#### 3.2.5. Mutilations

On mutilation procedures, allowing or denying tail docking, teeth clipping, and castration was reviewed. Routine tail docking is prohibited by the Council’s directive [2] but it is well known that, apart from Finland and Sweden, EU member states are in breach of this rule. Teeth clipping, the reduction of corner teeth, must not be performed routinely according to the directive [2]. The directive [2] allows surgical castration of male pigs by other means than tearing of the tissues and not later than at the seventh day of life. In organic pig production, tail docking is not allowed [8]. In addition, pain relief is required in connection with the castration of male piglets not later than the seventh day of its’ life [8]. With regards to teeth clipping, the guidelines for organic animal production follow the provisions of the directive [2].

BMEL’s AW label tier 1 allowed tail docking, which is below the minimum requirement of the council directive on the protection of pigs. It was also allowed by Beter Leven tier 1, the RSPCA, and HFAC in exceptional circumstances, and also by Bedre Dyrevelfærd tier 1 for “individual pigs if deemed necessary for veterinary reasons”. Likewise, BMEL’s label, the RSPCA, and HFAC did not allow teeth clipping routinely, but allowed it when needed. Teeth clipping was prohibited by Beter Leven, Coop DK, Friland, Naturaplan, Naturafarm and GAP. The information was not retrieved for Für Mehr Tierschutz and Bedre Dyrevelfærd.

Castration was allowed with certain conditions, under anesthesia and pain relief by most schemes, except in Beter Leven tier 1 and GAP tier 5 (Table 3). In RSPCA Assured, surgical castration was prohibited, but immunocastration allowed. Regarding the North American schemes, the additional conditions were related to the age of the pig. AWA and HAC allowed the surgical castration of pigs at a maximum of seven days of age and GAP before the age of 10 days (except tier 5, which prohibited the operation). Information was not available for Bedre Dyrevelfærd and Coop DK.

## 4. Discussion

In this study, 12 pig welfare labeling schemes were examined. The results suggest that there is a substantial variation between the labels in terms of which level of pig welfare they require and how the level is ensured. Variation between labels complicates their comparison, although similar factors are elaborated. In the communication of labeling schemes, the same animal welfare factors are promoted, but the level and requirements of these factors still varies. A more harmonized labeling approach, including harmonized terminology, may help to ensure that FAW is produced, communicated, and marketed in a more standardized manner, thereby ensuring that the animal welfare attributes of food of animal origin are easier to compare across labels in the EU’s common market.

The availability of information is one of the challenges, which limit the comparison. Many FAW assurances have been established but retrieving information from open web pages, especially in terms of FAW criteria, can be challenging. It is also an issue of trust and something that any label must overcome. The need for appropriate information of animal welfare has become increasingly important [45]. The European Union can be an active player in animal welfare issues, for example, at the World Organization for Animal Health. Besides, the level of animal welfare can be negotiated in the trade agreement between the EU and third countries. Thus, the EU has the potential to improve FAW globally if an EU animal welfare labeling scheme is introduced in the future [46]. It may also raise more consumer awareness to animal welfare related conditions and care outside the EU and Europe.

The more demanding tiers of the multitier schemes have the potential to improve animal welfare substantially. The highest tier of the multitier schemes often compares to organic standards providing outdoor facilities, additional space and comfort, and possibilities to behave according to the natural needs of pigs when compared to the minimum standards of the EU pig directive. However, despite the potential of the multitier scheme to allow farms to participate in the schemes with a lower level of ambition, from the FAW point of view, improvements can remain moderate in the low-ambition tiers. Sörensen & Schrader [15] have reviewed the major pig welfare problems that can be addressed in the criteria of the welfare labels, such as tail docking, floor quality, or a lack of enrichments. They concluded that there are several reasons why label schemes do not require significant criteria that improve animal welfare [15]. In addition to costs, there are also more complex reasonings, for example, the prohibition of farrowing crates can be seen to increase piglet mortality by sows crushing piglets. Studies indicate that although piglets are crushed in loose-housed systems, the total piglet mortality rate appears to be approximately at the same level in both well-managed loose housing and farrowing crate systems [47,48,49,50]. In general, many requirements, such as the amount of enrichments or bedding material, can be difficult to monitor consistently. This is an important aspect because an integral element of the schemes is the verification that the suggested welfare standards are met.

Resource-based measures are the dominant criteria in the FAW labels. Animal-based measures were used only seldomly as FAW labeling criteria in the reviewed schemes. Resource-based measures are comprehensive to consumers and are easier to check and communicate as they provide positive attributes. There can be challenges in promoting animal-based measures, such as reduced piglet mortality or skin damages, as they or their wording do not always represent a positive aspect of animal production, either from the consumer or industry perspective. However, the adoption of animal-based measures can improve FAW more than resource-based measures [15] and they will put the animal in focus. This suggests that more widespread use of animal-based measures in animal welfare labels will be warranted. Because animal welfare is an individual animal’s experience, the resources given to the animals do not solely guarantee the good welfare state of an animal. Observing the animal, whether it moves with its fellow animals, moves effortlessly, has scratches, patches, or injuries, or is either afraid of or curious about humans, reliably expresses the state of the animal’s experience [51,52].

However, resource-based measures are also important. For instance, allowing tail docking in an FAW scheme is questionable, as this is prohibited routinely in the EU by the Council Directive [2] upholding the minimum requirements for the protection of pigs. Tail biting can be reduced by improving housing conditions, e.g., by providing environmental enrichments [53] and exercise, which is an example of managing undesired behavior by the use of resource-based measures. Another example is the continuous availability to clean drinking water, which is a prerequisite for ensuring animals do not feel thirsty. Thirst is hard to observe and thus continuous access to clean drinking water can be considered as one of the necessary requirements for an animal welfare label. The advantage of resource-based measures is inspection in practice; they can be easier and less expensive to verify than animal-based measures. Credible inspections respond to consumer expectations for the ensured scheme to be trusted [14]. The Welfare Quality assessment method that has a wide scientific background applying animal-based measures can be utilized as a starting point for an animal welfare labeling scheme [54].

The animal welfare labeling schemes leave a lot of responsibility to consumers in estimating if FAW is improved. Different classifications based on the weight of the pig defining pigs’ minimum space requirement make the comparison of labels challenging. In addition, certain schemes define a minimum lying area while others do not. Hence, a claim such as “more space” may have label-specific meanings. When developing a labeling scheme, it is therefore important to attend to factors that make the scheme understandable, communicable, and interesting to the consumers such that they have an incentive to purchase labeled products. However, it is also important to ensure that the supply chain actors from farmers to retailers have incentives to participate in the scheme. Therefore, a multi-actor approach that engages different stakeholders in the development process is needed to ensure the adoption of the label.

An animal welfare label that is applicable in several countries will also raise possible differences in both welfare and farming conditions between the countries. It will facilitate the comparison of welfare across countries. While reviewed animal welfare schemes were heterogenous, flexibility may be necessary from the animal welfare perspective. For example, when discussing outdoor access, the climatic conditions are completely different across Europe. In Finland, the weather conditions during fall, winter, and spring, when the pastures are either covered with a thick layer of snow, or are wet and bare with no vegetation, challenge the conditions of outdoor facilities in terms of welfare. Another example is that in the Mediterranean, heat stress may be a more important welfare issue than in Northern Europe. There can also be tradeoffs between human and animal welfare regarding outdoor access and animal and zoonotic diseases. These can affect incentives to introduce welfare schemes. A difference can be seen between the focus of the food industry and consumer preferences for animal welfare. The most preferred measures by consumers are related to “naturalness” [14,55], outdoor access, additional space, and possibilities to express natural behavior. When developing a FAW labeling scheme, the improvement of the aforementioned resource-based characteristics should not compromise animal health and other sustainability dimensions, such as the environment or product safety.

## 5. Conclusions

FAW labels can ensure a certain level of welfare and communicate this to consumers. While the welfare requirements from the animal point of view are the same regardless of the country, there is considerable variation of the requirements between the FAW labeling schemes. The reviewed schemes showed similarities in terms of attributes and resource-based measures that were used to enhance animal welfare. Since the specific requirements are heterogeneous and different specifications can be applied, it varies by labeling system in how it can contribute to the welfare of farmed animals. The status of unlabeled animal production also affects how welfare labeled products can contribute to animal welfare in different countries. This complicates the comparison of labeling schemes. Harmonized terminology in FAW labeling and increased use of animal-based measures will facilitate more robust comparison of animal welfare labels. It is important to provide an open and coherent FAW labeling scheme from the consumer perspective because consumers are end users of welfare-friendly products and can pay for enhanced FAW. Adopting a harmonized labeling approach that regulates the technical standards (i.e., welfare criteria) and the terminology that can be used in communication would contribute to the availability of equivalent information and standardized animal welfare products across the EU.

## Figures and Tables

**Figure 1 animals-11-02430-f001:**
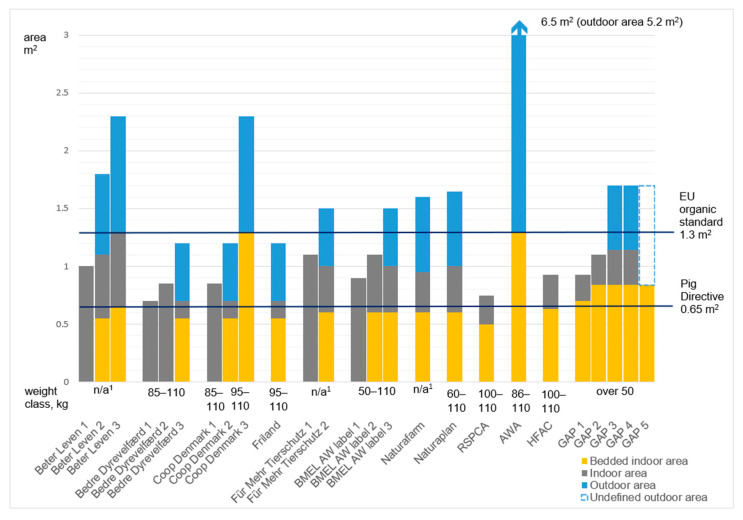
Minimum total area, lying area, and outdoor access requirement in the reviewed schemes. ^1^ n/a = not applicable.

**Figure 2 animals-11-02430-f002:**
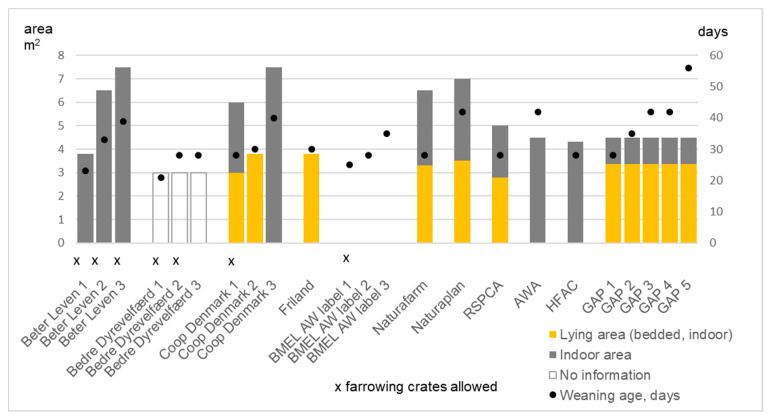
The size of farrowing place, the use of farrowing crates, and the minimum duration of suckling period, and hence the weaning age in 11 reviewed schemes. Label “Für Mehr Tierschutz” is not presented here as it did not consider farrowing issues other than prohibiting the use of the farrowing crate in tier 2. The EU pig directive [2] requires that piglets are not weaned before 21 days of age in the EU if they are moved into specialized housing. In organic production in the EU, piglets must not be weaned before 40 days of age [8].

**Table 1 animals-11-02430-t001:** Number of tiers, country of origin, establishment year, inspection frequency, and certification body of labels reviewed in the study.

Name	Number of Tiers	Country of Origin	Established	Inspection Frequency and Certification Body
Beter Leven	3	the Netherlands	2007	Annually, by external Beter Leven label foundation
Bedre Dyrevelfærd	3	Denmark	2017	Annually, by an accredited certification body
Coop Denmark	4	Denmark	2016	Annually, accredited by Baltic Control Ltd. according to EN 17065
Friland free range	1	Denmark	1992	Annually, by The Danish Animal Welfare Society inspector
Staatliches Tierwohlkennzeichen	3	Germany	to come	Twice each year, by an external body
Für Mehr Tierschutz	2	Germany	2009	At least two unannounced audits each year, risk-based control, and certification scheme, by an accredited certification body
Naturafarm Coop	1	Switzerland	2007	Regular unannounced inspections, by Swiss Animal Protection
Naturaplan Coop	1	Switzerland	1993	Annually, by an independent, state-accredited institutions
Royal Society for the Prevention of Cruelty to Animals (RSPCA) Assured	1	Great Britain	2015	Annually, by RSPCA Assured scheme assessors. In addition, at least 30% of farms receive a monitoring visit by an RSPCA farm livestock officer, often unannounced
Animal Welfare Approved	1	USA	2006 ^1^	Annually, by A Greener World’s independent trained auditors
Certified Humane	1	USA	1998	Annually, by inspectors who must have an MSc or PhD in animal science or a veterinary degree
Global Animal Partnership	5(+) ^2^	USA	2008	Certificate maintained for 15 months, by independent third party certifiers

^1^ Information based on the http://www.ecolabelindex.com/ecolabel/animal-welfare (accessed on 22 December 2020). ^2^ The scheme has 5 levels and an additional level 5+, where the transport of animals is prohibited, so the animals are slaughtered at the farm.

**Table 2 animals-11-02430-t002:** Enrichments, bedding, and floor type definitions in the animal welfare labeling schemes studied, the EU regulation on the protection of pigs (2008/120/EC), and in organic regulation (2007/834/EC).

Label and Tier Number	Enrichments	Bedding	Solid or Slatted Floor
Regulation on the protection of pigs 2008/120/EC	Permanent access to a sufficient quantity of material to enable proper investigation and manipulation activities, such as straw, hay, wood, sawdust, mushroom compost, peat, or a mixture of such	Bedding not requiredLying area must be comfortable, clean, and dry	Slatted floors allowed
EU organic regulation 2007/834/EC	Exercise areas shall permit dunging and rooting by porcine animals For the purposes of rooting, different substrates may be used	Bedding required, straw or other suitable natural material	A comfortable, clean, and dry laying or rest area of sufficient size, consisting of a solid construction which is not slatted
Beter Leven 1	e.g., edible material such as alfalfa in combination with a block of wood and/or a sturdy piece of rope ^1,2^	The solid floor part may consist of a rubber mat with litter on it	40% of the total area solid floor
Beter Leven 2	Ground cover such as straw or hay is extremely suitable	Straw or comparable material entire floor covered	50% of the total area solid
Beter Leven 3	Ground cover such as straw or hay is extremely suitable	Straw or comparable material entire floor covered	50% of the total area solid
BedreDyrevelfærd 1	Straw	Rooting material	Not applicable
Bedre Dyrevelfærd 2	Straw	Rooting material	Not applicable
Bedre Dyrevelfærd 3	Straw	Straw	Not applicable
Coop DK 1	Straw or roughage	Straw or roughage	Farrowing sows: 50% solid area
Coop DK 2	Straw, rough feed, roughage	Straw or wood shaving	
Coop DK 3	Straw	Straw	50% of the total area solid
Friland	No information available	Straw, flakes, or similar material	Slatted floor not allowed at the lying areas. Only 50% of non-bedded area may be slatted.
Für Mehr Tierschutz 1	Organic material	No requirements ^3^	0% (no requirement for solid area)
Für Mehr Tierschutz 2	Organic material	Straw or some other bedding material	Lying area must be solid
BMEL FAW label 1	Materials such as straw, hay, sawdust, mushroom compost, or peat	Not applicable	Lying area must be solid
BMEL FAW label 2	Materials such as straw, hay, sawdust, mushroom compost, or peat	The lying area must be bedded, no requirements for bedding material	Lying area must be solid
BMEL FAW label 3	Materials such as straw, hay, sawdust, mushroom compost, or peat	The lying area must be bedded, no requirements for bedding material See the organic label requirements	Lying area must be solid
Naturafarm Coop	Straw, different surfaces	Bottom-covered, dry-lying area without perforation	50% of total area solid
Naturaplan Coop	Straw, different surfaces	All lying areas must have bedding	Lying area must be solid
RSPCA Assured	High quality straw, peat, and silages	Bedded to a sufficient extent to avoid discomfort, solid construction	Lying area must be solid
AWA	Where vegetative cover cannot be maintained throughout the year manipulable material must be provided	Recommended bedding with straw or corn stover is preferred	100% solid floor
HFAC	e.g., wood chips, sawdust, or peat Also objects for manipulation, such as chains, balls and materials such rope	Bedded to a sufficient extent to avoid discomfort	Lying area must be solid
GAP 1	No requirements	e.g., sawdust, wood shavings, wood chips, rice hulls, long or chopped straw, alfalfa pellets, and corn stalks	75% of the total area solid
GAP 2	e.g., long straw, hay, silage, wood chips, branches, whole crop peas or barley, compost, peat, sisal ropes, or other natural material	e.g., sawdust, wood shavings, wood chips, rice hulls, long or chopped straw, alfalfa pellets, and corn stalks	75% of the total area solid
GAP 3	e.g., long straw, hay, silage, wood chips, branches, whole crop peas or barley, compost, peat, sisal ropes, or other natural material	e.g., sawdust, wood shavings, wood chips, rice hulls, long or chopped straw, alfalfa pellets, and corn stalks	75% of the total area solid
GAP 4	e.g., long straw, hay, silage, wood chips, branches, whole crop peas or barley, compost, peat, sisal ropes, or other natural material	e.g., sawdust, wood shavings, wood chips, rice hulls, long or chopped straw, alfalfa pellets, and corn stalks	75% of the total area solid
GAP 5	Pasture access at all times		

^1^ Straw is not required for groups of less than 40 pigs [17]. ^2^ The enrichment material must be accessible to 25–50% of the animals at the same time [17]. ^3^ Every pig keeper has to test the comfort of lying areas in at least four pens.

**Table 3 animals-11-02430-t003:** Criteria for male pig castration in reviewed animal welfare labels.

Label and Tier	Castration Policy
Beter Leven 1	Not allowed
Beter Leven 2–3	Allowed surgically with anesthesia, pain medication required afterwards
Bedre Dyrevelfærd 1–3	Not applicable
Coop Denmark 1–3	Not applicable
Friland	Allowed surgically with anesthesia, pain medication required
Für Mehr Tierschutz 1–2	Allowed surgically under general anesthesia (isoflurane), pain medication required, immunological castration allowed
BMEL AW label 1–3	Allowed surgically with anesthesia
Naturafarm Coop	Allowed surgically with anesthesia
Naturaplan Coop	Allowed surgically with anesthesia
RSPCA Assured	Only immunological castration allowed
AWA	Allowed surgically for up to 7-day-old pigs, immunological castration prohibited
HFAC	Allowed surgically for up to 7-day-old pigs
GAP 1–4	Allowed surgically for up to 10-day-old pigs
GAP 5	Not allowed

## Data Availability

Data supporting reported results can be enquired from the first author.

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
