# Peer review of "Comparison of 12 Different Animal Welfare Labeling Schemes in the Pig Sector"

_animals, 2021, doi:10.3390/ani11082430_

Round 1

Reviewer 1 Report

Manuscript Number: animals-1305180-peer-review-v1

Title: Comparison of 12 different animal welfare labeling schemes in the pig sector

Authors: Katriina Heinola, Tiina Kauppinen, Jarkko K Niemi, Essi Wallenius and Satu Raussi

Overview and general recommendation:

The aim of the manuscript was to analyse the characteristics and comparison of selected farm animal welfare (FAW) labelling schemes for pigs. The authors compared 12 selected FAW labelling schemes and additionally they referred to the minimum standards of the EU requirements for the protection of pigs and of organic production. The research subject is very actual and interesting and manuscript itself is well written. In my opinion manuscript is valuable and  can be an important voice in the discussion on the need to develop a coherent, international system of animal welfare assessment. However, before the manuscript will be considered for the publication in Animals some of the points should be analysed and corrected.

Major comments:

In several places in the manuscript, authors refer to the consumer perception of the labeling products with information about animal welfare. It should be noted, however, that the authors did not investigate this (e.g. questionnaire research was not carried out or not described in the manuscript, how the issue of animal welfare labelling schemes is assessed from consumer perspective (e.g. access to information on animal welfare, readability / comprehensibility of information, assessment of the need to introduce labelling, the impact of labelling on making purchasing decisions). In my opinion, these elements should be removed from the manuscript. Among others the above texts will require redrafting / deletion: “In addition, this study examined how a consumer can access information about the labelling schemes and understand the added value of labeling for pig welfare.” as well as “The results suggest that it is challenging for con-sumers to compare existing FAW labelling schemes and that there is room for a harmonized labelling scheme between at least the EU countries, harmonized terminology in FAW labelling and increased use of animal-based measures in labels”.

The authors present the results in the form of Figures: 1 and 2 as well as Tables: 1 and 2. However, some of the results are neither presented in tables nor in graphical form but are only described in the text. In my opinion, in order to increase the transparency of the presentation of results, it is worth trying to present the other results in the form of a tables (e.g using the terms "yes / no" or "+ / -" for a given requirement).

Minor comments:

Throughout the manuscript, in many places, words are separated by dashes, these should be corrected.

In my opinion, in chapter Introduction table 1 (which contains the information that is the subject of the article) should not be quoted. The above text should therefore be edited or deleted: „For instance, two different labels were launched in Denmark within a short time period, ‘Animal welfare hearts’ by Coop retailer in 2016, and Bedre Dyrevelferd by the Danish Gov-ernment in 2017 (see Table 1)”

The next section presents labelling schemes and attributes studied, while subsequent sec-tions analyze and discuss about the characteris-tics of the schemes.” - incomprehensible text / should be rewritten or deleted.

In chapter “3.1. Inspection of the schemes” the Table 1 should be quoted.

In chapter 3.2.2. Floor type, bedding and enrichments: “The examples of enrichments and bedding type are listed in ap-pendix (table 3)” – please remove the reference of table 3 (it might be misleading what table is meant).

In chapter „3.2.3. Farrowing-related criteria” please add reference for the sentence: “The EU pig directive defines that production no piglets must not be weaned before 28 days, but there are exceptions for the shorter period, only 21 days, when piglets are moved to specialized housing”.

Figure 2. Weaning age should not be presented as a line connecting different systems (there can be bars - but there cannot be a continuous line).

Figure 2. In the footnote under Figure please add information why there is 11 not 12 schemes.

Figure 2. If it is possible, then, as in Figure 1, please add reference to EU regulations.

In chapter 4. – please identify to what „quality” refers to in sentence: “The common label in the EU could help to harmonize the varying criteria and thus guarantee the quality.” The sentence should be reworded or supplemented.

The last paragraph of the Chapter 4. Discussion might be separated as “Conclusions”. Without these separation the information: “…the welfare requirements from the animal point of view are the same regardless the country, but there is a lot of variation between the requirements of the AW schemes” is a repetition of previous statements. However if it will be in a separate "Conclusions" section, it won't be a problem.

Reviewer 2 Report

In the recent years, public has been increasingly concerned about animal welfare. It is true especially in western countries where consumers intentionally search for animal-friendly products and are willing to pay more if they believe they support a better life of farm animals. However, orientation in the market is not easy when they cannot directly buy from the local farmers they know in person. Animal welfare labelling schemes are supposed to help but as mentioned by the authors differing requirements of animal welfare schemes make it hard for consumers to make informed choices too. The authors compared 12 different animal welfare labeling schemes in the pig sector and as expected they found major differences among them and thus the respective level of animal welfare they can quarantee. No doubt a harmonized labelling scheme would be welcome but there is no suggestion how to achieve that.

Specific comments:

General

The manuscript should be thoroughly checked and corrected - many words are incorrectly hyphenated making the text difficult to read.

The statement "the welfare requirements from the animal point of view are the same regardless the country" is repeated way too many times.

Methods

Not clear whether the authors made a selection among the animal welfare label schemes found when searching for existing schemes (if it is the case please describe the selection process) or all animal welfare label schemes they had found were included in the analysis. Please clarify.

Conclusion

The hypothesis that the different animal welfare labeling schemes compared in the study vary a lot was (not surprisingly) confirmed. An adoption of a harmonized labelling scheme is suggested. What steps do you propose to take to achieve that?
